# Measuring supply-side service disruption: a systematic review of the methods for measuring disruption in the context of maternal and newborn health services in low and middle-income settings

Catherine R McGowan  ,[1] Dhikshitha Gokulakrishnan,[2] Emily Monaghan,[3,4] Nada Abdelmagid,[1] Laura Romig,[5] Meghan C Gallagher ,[5] Janet Meyers,[5] Rachael Cummings,[3,6] Laura J Cardinal[3]

For numbered affiliations see end of article.

**Correspondence to**
Dr Catherine R McGowan;
Catherine.McGowan@lshtm.ac.uk

## ABSTRACT

**Objectives** During the COVID-19 pandemic, most essential services experienced some level of disruption. Disruption in LMICs was more severe than in HICs. Early reports suggested that services for maternal and newborn health were disproportionately affected, raising concerns about health equity. Most disruption indicators measure demand-side disruption, or they conflate demand-side and supply-side disruption. There is currently no published guidance on measuring supply-side disruption. The primary objective of this review was to identify methods and approaches used to measure supply-side service disruptions to maternal and newborn health services in the context of COVID-19.

**Design** We carried out a systematic review and have created a typology of measurement methods and approaches using narrative synthesis.

**Data sources** We searched MEDLINE, EMBASE and Global Health in January 2023. We also searched the grey literature.

**Eligibility criteria** We included empirical studies describing the measurement of supply-side service disruption of maternal and newborn health services in LMICs in the context of COVID-19.

**Data extraction and synthesis** We extracted the aim, method(s), setting, and study outcome(s) from included studies. We synthesised findings by type of measure (ie, provision or quality of services) and methodological approach (ie, qualitative or quantitative).

**Results** We identified 28 studies describing 5 approaches to measuring supply-side disruption: (1) cross-sectional surveys of the nature and experience of supply-side disruption, (2) surveys to measure temporal changes in service provision or quality, (3) surveys to create composite disruption scores, (4) surveys of service users to measure receipt of services, and (5) clinical observation of the provision and quality of services.

**Conclusion** Our review identified methods and approaches for measuring supply-side service disruption of maternal and newborn health services. These indicators

## STRENGTHS AND LIMITATIONS OF THIS STUDY

⇒ This study presents various methods and approaches used to measure supply-side disruption in the context of maternal and newborn health; despite their various limitations, these methods and approaches could be incorporated into preparedness and response guidance for measuring supply-side service disruption and for early mitigation against the effects of disruption on essential services.

⇒ Despite the absence of comprehensive guidelines for measuring supply-side service disruption, we were able to identify several studies describing methods and approaches used to measure COVID-19-related disruption to maternal and newborn health services in low-income and middle-income settings.

⇒ This study also presents a summary of qualitative findings about the experience of service disruption; these findings may be used to inform the design of quantitative indicators of supply-side service disruption.

⇒ We did not carry out an assessment of study quality, nor did we assess risk of bias; though this was appropriate given the aim to create a typology of quantitative indicators, this approach may have introduced low quality and/or unreliable qualitative findings.

provide important information about the causes and extent of supply-side disruption and provide a useful starting point for developing specific guidance on the measurement of service disruption in LMICs.

## INTRODUCTION

With few exceptions, the COVID-19 pandemic exposed the vulnerabilities of health systems globally. In response to the burden of COVID-19 cases, many countries closed or

scaled-back non-essential health services to allow relocation of resources to the COVID-19 response, and to reduce health system strain. In mid-2020, the WHO conducted the first of three key informant surveys among national health authorities in 105 countries.[1] These 'pulse surveys' aimed to assess the impact of COVID-19 on 25 essential health services. Nearly all countries reported service disruptions, with lower-income countries reporting greater service disruption than higher-income countries (HICs). The causes of disruption were identified as a mixture of service supply and service demand factors. The most reported supply-side cause of disruption was the cancellation of elective services (reported by 66% of countries). Other supply-side causes included: staff redeployment to COVID-19 response (49%) or insufficient staff to provide services (29%), insufficient personal protective equipment for healthcare workers (44%), closure of health services (33%) or facilities (41%), and interruptions in the availability of medical supplies (30%).[1] Demand-side indicators may function as proxy indicators of supply insofar as they may evidence a reluctance to present to health services, or challenges with access. Demand-side indicators may demonstrate reduced uptake despite continuity, or scale up, of services.

Defining a package of essential services is a key preparedness activity; however, scaling back non-essential services may not be sufficient to avoid widespread and lasting disruption. The latest WHO pulse survey (published May 2023) notes that despite evidence of health system recovery, 84% of participating countries reported ongoing disruption to essential health services in the last quarter of 2022,[2] suggesting that disruption persists well into the recovery period. In June 2020, the WHO published operational guidance for maintaining essential health services during the pandemic.[3] The guidance cautions that all health system adaptations, '…should be made in accordance with ethical principles, such as equity in the allocation of resources and access, self-determination, non-abandonment and respect for dignity and human rights' (WHO,[3] p03). Though every country determines its own essential health services package based on its local contexts and capacities, the WHO recommended that seven health service categories be maintained during the acute phase of the pandemic including: (1) prevention and treatment of communicable diseases, (2) reproductive health services (including obstetric and child health), (3) core services for vulnerable populations (including infants and older adults), (4) chronic diseases (including mental health conditions), (5) critical facility-based therapies, (6) management of emergency health conditions, and (7) auxiliary services (including diagnostic imaging and laboratory services). The WHO guidance emphasised that critical decisions about the nature and timing of modifications to service delivery, '… must be informed by the use of accurate and timely data throughout all phases of the COVID-19 pandemic' (WHO,[3] p18).

Data on service disruption should be suitable to inform rapid and appropriate adjustments (eg, workforce optimisation strategies, resource allocation adaptations), including adjustments across essential services to ensure health equity.[4] While supply-side disruptions may contribute to decreased uptake (and vice versa), defining indicators which isolate the direction of disruption can potentially inform measurements of the extent of demand-side disruption that can be attributed to supply-side disruption. In addition, monitoring essential health services requires: meaningfully disaggregated data (to ensure that services are being delivered equitably), data that allow comparison of disruption over time (eg, across key phases of the pandemic), and rapid needs assessments to evidence facility-level priorities.[3 5] Both the WHO guidance on maintaining essential health services, and the guidance on using routine data to monitor the effects of COVID-19 on essential services, include lists of sample indicators for measuring disruption; however, most of these indicators measure demand-side disruption as a proxy for supply-side disruption (eg, number of women presenting to facility), or as a composite measure of both supply-side and demand-side disruption (eg, number of facility births).[4 6]

## Disruption of maternal and newborn health services during COVID-19

The negative impact of the COVID-19 pandemic on maternal and newborn health (MNH) outcomes has been well established at the population level. However, the causal pathways that create poor outcomes are less well understood.[7] The evidence suggesting a direct link between maternal COVID-19 infection and maternal/newborn outcomes is limited.[8 9] It is suspected that disruption to MNH services—either in terms of reduced overall provision of services, or reduced quality of services—was primarily driving poor MNH outcomes.[10] It has been estimated that supply and demand side disruption, combined with increasing food insecurity, could lead to between 12 200 and 56 700 excess maternal deaths globally.[11] Vulnerable populations, including women and children, are experiencing the greatest effects (both direct and indirect) of the pandemic, thereby widening existing inequities[12] and reversing progress towards improving the health of these and other vulnerable groups.[7 10]

Though nearly all countries reported disruptions to essential health services, disruptions were often more severe in low and middle-income countries (LMICs).[13] While all essential health services have been disrupted by COVID-19, some have been disproportionately affected.[13] Evidence suggests that MNH services have been disproportionately impacted by COVID-19-related service disruption, particularly in LMIC settings.[14 15] A scoping review of the impact of COVID-19 on maternal, newborn, child health and nutrition concluded that a considerable amount of funding for these services in fragile and conflict-affected settings had been diverted to COVID-19 response activities.[16] A recent systematic review aiming to synthesise the direct and indirect effects of the pandemic concluded that, '…minor consideration was given to

preserving and promoting health service access and utilisation for mothers and children, especially in historically underserved areas in Africa' (Adu *et al*,[17] p1).

## Rationale

The need for better service coverage data was identified early in the pandemic as the limitations of routinely collected data for reporting disruption became apparent.[4] Currently, both the qualitative and quantitative literature over-represents demand-side disruptions. Moreover, many crude health outcome indicators, such as number of stillbirths and maternal mortality, conflate supply-side and demand-side disruptions. Relatively few reports of COVID-19-related service disruption aim to isolate supply-side disruption, yet the measuring of supply side disruptions is key to mitigating their effects. While we were able to identify useful guidance on monitoring the effects of COVID-19 on essential health services, this guidance focused exclusively on routinely collected data.[6] Importantly, the authors caution that routine health information systems (RHISs), '…present an incomplete picture of the services used by communities because reporting rates are often low and many private care facilities (including those run by nongovernmental or religious organisations) do not report data to RHIS' (WHO,[6] p5). In addition, RHISs present only a partial indication of the impact of COVID-19 on essential services in settings in which a significant proportion of health services are provided through the private sector.[6] Finally, RHIS data vary in terms of data quality and completeness, and population denominators are particularly prone to error. This review aims to address existing gaps by describing how COVID-19-related, supply-side disruption has been measured in the context of MNH services in LMICs during COVID-19. In addition, the design of indicators is typically evidenced by qualitative research findings; as such, we sought to consolidate qualitative findings regarding the nature and extent of service disruption.

## Objectives

The objectives of this review are: (1) to present methods and approaches used to quantitatively measure supply-side service disruptions in LMICs in the context of COVID-19 and MNH services, and (2) to consolidate the qualitative evidence of supply-side service and quality disruptions to MNH services in LMICs.

This review is reported against the Preferred Reporting Items for Systematic Reviews and Meta-Analyses (PRISMA) 2020 checklist[18] and updated reporting guidance.[19] The review protocol has been published in the PROSPERO prospective systematic review registry (CRD42022381537).[20]

## METHODS

### Eligibility criteria

Eligibility was defined as any empirical source that reported on at least one indicator of supply-side disruption, regardless of method (ie, quantitative or qualitative). We defined supply-side disruption broadly to include cancelling/scaling back of an MNH service, or reduction in quality. We included studies reporting on any specific MNH clinical service (eg, assisted delivery, antenatal care (ANC), postnatal care (PNC) for mothers and newborns, essential newborn care), or MNH services more broadly (even if their components were not described). We also included studies of multiple essential services disaggregated by service and including at least one MNH service. We define supply-side disruption broadly to include disruption to the provision of care (eg, mean difference in the number of health workers in the labour room), and disruption that compromises quality of care (eg, provision of timely and appropriate care, standard precautions for preventing hospital-acquired infections followed). We did not include in either category perceived disruptions to quality of care attributed to minimal and reasonable infection prevention and control requirements (eg, the requirement to wear masks during consultation, social distancing while in a queue).

We did not include: reports of disruption solely in the context of family planning or sexual health services; studies reporting only on MNH service uptake (or on indicators which clearly combine, or otherwise conflate, demand side and supply side disruptions); or comparisons of baseline (eg, pre-COVID-19) with follow-up outcome indicators (eg, stillbirths) that did not isolate the effect of supply-side disruption. Sources were also excluded if they reported on supply chain disruption (eg, blood for transfusion) without reporting on resulting discontinuation, scale-down, or reduced quality of services. We included sources written in any language. Finally, we excluded preprints.

### Information sources and search strategy

We searched the MEDLINE, EMBASE and Global Health databases (via Ovid) for peer-reviewed and relevant grey literature on 23 January 2023; no start date was used as this was obviated by the inclusion of COVID-19 as a search term. We adapted our search strategy from the search methods used to maintain and update the Cochrane Pregnancy and Childbirth Group's Specialised Register.[21] The complete database search strategy is included in online supplemental appendix 1. The grey literature searches were carried out via DuckDuckGo (23 February 2023) and Google Search (26 February 2023) using the same search strategy for both search engines. Search terms included: '*service disruption*' and *COVID*, '*essential service disruption*', *maternal* and *newborn* and '*service disruption*', *maternal* and *newborn* and *services* and *COVID*. The team also searched relevant websites including the WHO Global Pulse Survey website[22] and

the WHO COVID-19 Technical Guidance for Maintaining Essential Services and Systems website.[23]

## Selection process, data collection and synthesis

The lead author (CRM) carried out the initial screening of all sources retrieved via database searching. Screening of grey literature sources was carried out by two authors (EM/LR). The lead author manually extracted all data in NVivo V.1.0 (Melbourne, Australia: QSR International) based on the following characteristics and domains: title, year, author, aim, method, setting, time (data collection), programme or population, findings, and definition of disruption. We compiled a narrative synthesis of the literature. Only data on supply-side disruption were extracted from sources that reported on both demand-side and supply-side factors. We defined *quality of service* in line with WHO definition of quality (ie, the service is effective, safe, people-centred, timely, equitable, integrated and efficient) and the WHO standards for improving the quality of maternal and newborn care.[24 25] We defined *provision of service* to include cessation or scale back of services.

## Reporting bias and certainty assessment

We did not assess bias of the studies themselves. While we had intended to use an established guideline to evaluate the quantitative indicators identified in the literature, no source provided sufficient information to allow a complete quality assessment.

## Patient and public involvement

None.

# RESULTS
## Study selection

The initial database search returned 4438 records published on or before 23 January 2023. We used EndNote V.20.5 (Philadelphia, Pennsylvania, USA: Clarivate) to delete duplicates which resulted in 2657 unique records. Initial screening was carried out based on review of the title/abstract and resulted in 94 sources for which we were able to retrieve 85 full-text documents. Twenty-eight peer-reviewed sources remained following full-text review. The PRISMA flow chart is included in figure 1.

The reasons for exclusion, following full-text review, were: only reported on demand-side disruption (n=17), not sufficiently focused on disruption (n=18), not sufficiently focused on MNH (n=5), not sufficiently focused on LMICs (n=2), review article (n=3), not empirical (n=2), no disruption (n=3), preprint (n=6) and evaluation of a particular MNH intervention (n=1). The grey literature search produced 21 unique results. None of the grey literature sources met the inclusion criteria as: they were not empirical (n=9), were not focused on MNH

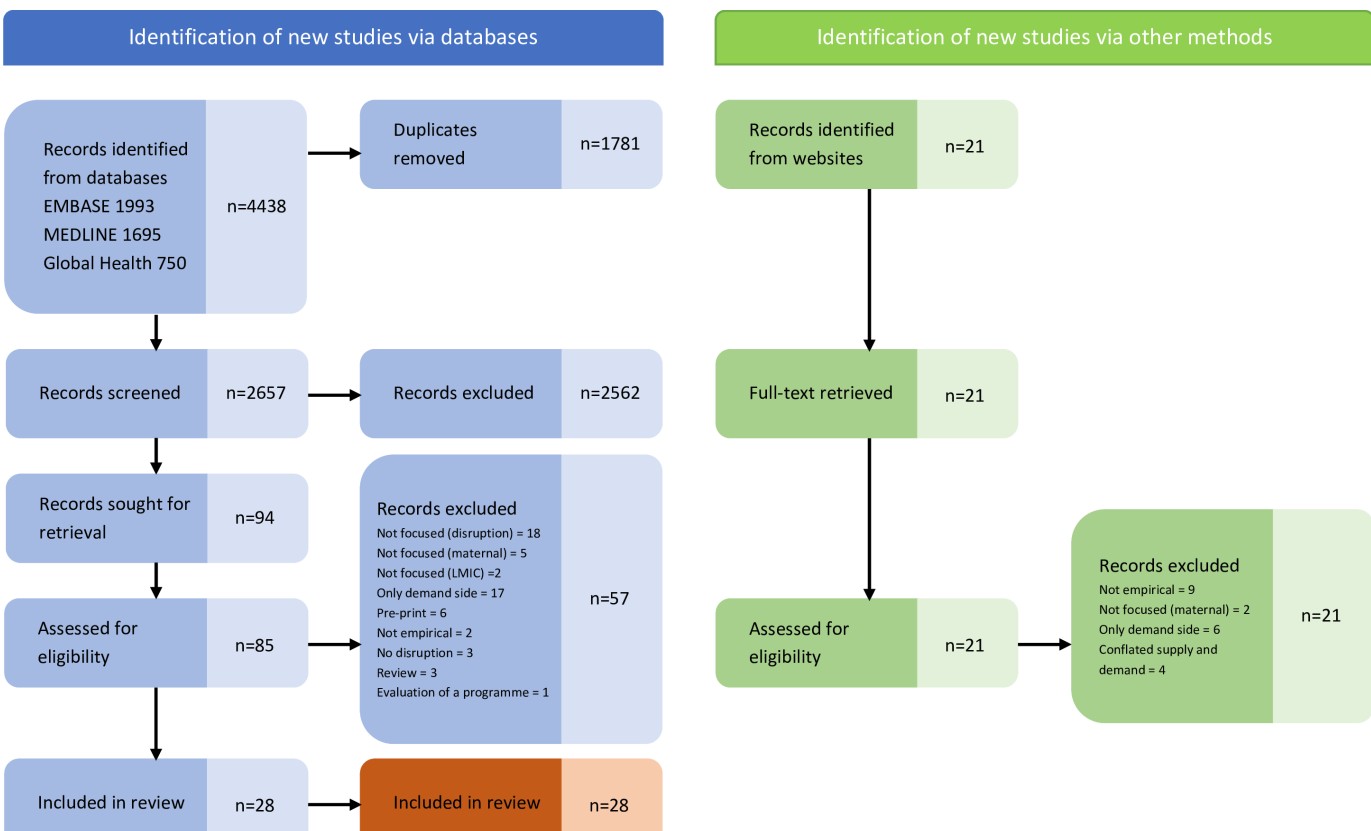

**Figure 1** PRISMA flow chart. LMIC, low-income and middle-income country. PRISMA, Preferred Reporting Items for Systematic Reviews and Meta-Analyses.

(n=2), presented only demand-side indicators (n=6), or they included indicators of disruption which did not isolate supply-side factors (n=4).

## Study characteristics

The characteristics of the 28 included studies are listed in online supplemental appendix 2. Studies were carried out in: Bangladesh,[26] Ethiopia,[27 28] India,[29–34] Indonesia,[35 36] Iran,[37 38] Nepal,[39 40] Nigeria[41–44], and Uganda.[45 46] Three studies included multiple LMIC countries,[14 47 48] and four included both HICs and LMICs.[49–52] Most studies (n=18) used qualitative methods to collect data about supply-side disruption[28 30 31 33–37 40–47 49 50]; one study involved a qualitative media analysis.[32] Seven studies were based on quantitative methods,[14 26 27 29 38 48 52] and two used both qualitative and quantitative methods.[39 51] Most studies (n=14) involved participation from service-providers (eg, nurses, midwives, obstetricians, community health workers, policy-makers),[14 29 30 35 40 42–44 47–52] while 11 involved service users,[27 31–34 36–39 45 46] and 3 included both groups.[26 28 41]

## Results of synthesis

We report below a synthesis of the quantitative approaches used to measure supply-side disruptions to the provision of services and/or the quality of services due to COVID-19—a summary of these findings can be found in tables 1 and 2. In addition, we summarise the qualitative approaches to measuring supply-side disruption.

### Quantitative measures of disruption

Quantitative indicator data were collected using the following approaches:

#### Surveys of healthcare workers or health stakeholders about disruptions or missed services

Surveys of healthcare workers or health stakeholders about disruptions[48 51 52] or missed services[27] were carried out in four studies. The first involved distributing a survey among healthcare workers in 51 countries (including 32 LMICs)—asking if they were aware of any problems providing MNH immunisation services (yes/no)—if respondents answered yes, they were asked to indicate the

---

**Table 1** Measures of disruption

| | Provision of services | Quality of services |
|---|---|---|
| Qualitative methods | ► Semistructured interviews with healthcare workers and/or service users asking about difficulties providing care.[28 30 31 33 34 36 37 39 40 42–46]<br>► Free-text field in a survey.[50 51]<br>► Semistructured interviews with healthcare workers asking how they provided care at T1 (retrospective baseline) and how they currently (T2) provide care.[35]<br>► Semistructured interviews with healthcare workers which ask about impacts at various levels of the health system.[47]<br>► Review of news articles reporting on disruption.[32] | ► Semistructured interviews with health staff and/or service users.[28 30 33 37 39–41 43 44 46 50 51]<br>► Surveys of health workers which include a free-text field asking to what extent has care been provided at T1 (retrospective baseline) compared with T2.[49]<br>► Semistructured interviews with healthcare workers which ask about impacts at various levels of the health system.[47]<br>► Review of news articles reporting on disruption.[32] |
| Quantitative methods | ► Observation of health facilities (to determine no of healthcare workers on wards, days service disrupted).[39]<br>► Surveys of healthcare workers which ask respondents to provide a 'disruption score'.[14]<br>► Surveys of healthcare workers which ask about service provision at T1 (retrospective baseline) and T2.[29]<br>► Surveys of healthcare workers, asking about service provision, taken at baseline (T1) and repeated at T2 and T3.[26]<br>► Surveys of healthcare workers about disruptions.[48 51 52]<br>► Surveys of service users (via SMS) asking them to indicate if they received the health service for which they attended the facility.[38]<br>► Survey of service users asking them if they missed a healthcare service and if yes then they are asked to indicate why from a list of facility-related factors.[27] | ► Surveys of healthcare workers, asking about service quality, taken at baseline (T1) and repeated at T2 and T3.[26]<br>► Surveys of healthcare workers about disruptions.[52] |

---

**Table 2** Indicators

| Author | Location | Design | Measure | Population | Intervention | Indicator | Example |
|---|---|---|---|---|---|---|---|
| Ashish[39] | Nepal | Observation of clinical services | Extent of disruption | Service-providers | Maternity services | Measured the mean difference of the no of health workers in the labour room and the no of days that maternity services were disrupted between the two time periods (prepandemic and pandemic). | For example, mean no of days (4.3) that maternity services were halted during the pandemic and concluded that the no of healthcare workers (per 24 hours in the labour and delivery room) had decreased to 5.4 (from 6.2 before the pandemic) health workers due to redeployment to COVID-19 dedicated care. |
| Assefa[14] | Burkina Faso, Ethiopia, Nigeria | Cross-sectional survey | Extent of disruption | Service-providers | General (includes iron and folic acid, ANC, and family planning services) | Measure of the subjective assessment of the categorical level of service disruption presented as a mean interruption score. | For example, maternal and reproductive services were scored an average of 2.24 out of 6 (6=total disruption) for Nigeria, 1.72 out of 6 for Burkina Faso and 1.67 out of 6 for Ethiopia |
| Avula[29] | India | Cross-sectional survey (asking about T1 and T2) and routine HIS data on service provision (T0) | Extent of disruption | Service providers | Iron and folic acid and ANC | Measure of changes in service provision (categorically defined at T1 and T2 as proportion of women provided with ANC services, and at T0 as the no of women receiving services) across the prepandemic, lockdown and post-lockdown periods. | For example, ANC services were disrupted during the lockdown period but were restored during the post-lockdown period (but not to prepandemic levels). |
| Nguyen[26] | Bangladesh | Longitudinal survey (at T0, and repeated at T2 (at which time participants asked to recall services provided in the month preceding the survey (T2) and during lockdown (T1)) | Extent of disruption | Service users and service providers | Iron and folic acid and ANC/PNC | Proportionate measure of service provision (defined as the proportion of service providers indicating provision of specific services, and the proportion of service users indicating receipt of specific services) at pre-pandemic (T0), lockdown (T1) and post-lockdown (T2). | For example, fewer facilities offered ANC services during lockdown when compared with before the pandemic (dropping 6.6 percentage points), and that there were significant drops in the receipt of anthropometric measurements, and iron/folic acid/calcium supplementation. All services recovered post-lockdown but not to the prepandemic levels. |
| Rezapour[38] | Iran | Survey | Extent of disruption | Service users | General (includes midwifery services) | Proportionate measure of services received as a factor of total presentations, by month. | For example, the study found that there was a small but significant drop in the APSD for midwifery services when comparing the pandemic to the pre-pandemic periods. |
| Saso[51] | 51 countries | Cross-sectional survey | Extent of disruption | Service providers | Maternal and infant vaccination services | Proportionate measure of subjective assessment of disruption to vaccine delivery across several countries. | For example, 53% of LMIC-based respondents reported disruptions to maternal immunisation services compared with 42% of HIC-based respondents. |
| Semaan[52] | 61 countries | Cross-sectional survey | Causes of disruption | Service providers | Maternal and newborn PNC | Proportionate measures of the subjective assessment of the causes of disruption (in terms of the provision and quality of services) to PNC across multiple countries. | For example, respondents from LMICs reported various disruptions due to COVID-19 including reduced numbers of beds due to social distancing (37%), cessation of home visits (20%), and suspension of PNC (8%). |

Continued

**Table 2** Continued

| Author | Location | Design | Measure | Population | Intervention | Indicator | Example |
|---|---|---|---|---|---|---|---|
| Tadesse[27] | Ethiopia | Cross-sectional survey | Causes of disruption | Service users | ANC services | Proportionate measure of disrupted (delayed or missed) ANC visit as well as a proportionate subjective assessment of the causes of the disruption. | For example, 216 respondents reported missed or late ANC visits, of which 72 (33.33%) attributed this to diversion of maternal services to COVID-19. |
| Villalobos Dintrans[48] | Latin America and Caribbean | Cross-sectional survey | Extent of disruption | Service providers | General (includes MNH services) | Proportionate measure (using a Likert scale: suspended, partially reduced, same as before, adapted to circumstances, new service created) of the perception of the impact of COVID-19 on health during the pandemic. | For example, services for newborns and pregnant women —such as institutional delivery care and postpartum care— as well as vaccination programmes, showed the best perceptions of coverage during the pandemic. |

ANC, antenatal care; APSD, actual percentage of service delivery; HIS, health information system; MNH, maternal and newborn health; PNC, postnatal care.

nature of the problems (described below in the section on qualitative measures).[51] This approach produced a crude proportionate measure of the subjective assessment of disruption to vaccine delivery across several countries (eg, 53% of LMIC-based respondents reported disruptions to maternal immunisation services compared with 42% of HIC-based respondents).

Using a predefined list of disruptions to the provision and quality of services, a study in 61 countries (including 34 LMICs) asked healthcare workers to indicate how specific MNH services (eg, inpatient/outpatient PNC) were affected by COVID-19 in the preceding month.[52] This study produced proportionate measures of the subjective assessment of the causes of disruption to PNC across multiple countries; for example, respondents from LMICs reported various disruptions due to COVID-19 including reduced numbers of beds due to social distancing (37%), cessation of home visits (20%), and suspension of PNC (8%).[52]

A study among health stakeholders in 19 Latin American and Caribbean countries asked respondents to indicate their perception of the disruption to various health services (including MNH) during the pandemic using a Likert scale (ie, services have been maintained, partially reduced, suspended, adapted to COVID-19, and new services were created).[48] The study concluded that services for newborns and pregnant women (eg, institutional delivery care, postpartum care, vaccination programmes) were found to have showed the best perceptions of coverage during the pandemic.[48]

Finally, a survey to assess disruption to ANC services was sent to pregnant women in Ethiopia asking, 'Did you miss or were late to start an ANC visit during the COVID-19 outbreak?' (yes/no).[27] Respondents were asked to indicate a reason for missed or delayed ANC visits from a selection of responses describing facility-related factors for disruption; such as, 'deploying of maternal care workers', 'interruption and diversion of maternity services to the COVID-19 response', 'fear of COVID-19 infection', and 'social distancing'. The study produced a measure of the proportion of respondents who experienced a disrupted ANC visit (which does not isolate supply-side factors) as well as a proportionate subjective assessment of the causes of the disruption. Two-hundred and sixteen (out of 389) respondents reported missed or late ANC visits, of which 72 (33.33%) attributed this to diversion of maternal services to COVID-19.[27]

### Surveys of healthcare workers or health stakeholders to measure temproal changes in disruption

Surveys of healthcare workers or health stakeholders (to measure temporal changes in disruption) were used in two ways: (1) in India retrospective baseline data were compared to accounts of the situation at the time of the interview,[29] and (2) in Bangladesh, data about service provision and service quality were collected from healthcare workers and mothers with children under two at baseline (ie, time 1 (T1)), and at T3 (which included retrospective questions about T2).[26]

The study from India involved sending a survey to healthcare workers asking if they were able to provide ANC services to all pregnant women during the most recent lockdown (T1), and in the preceding month (T2). Responses were: 0 (none), 1 (yes, but only to some pregnant women) or 2 (yes, to all pregnant women). Data about prepandemic service provision (T0) were collected from hospital HIS systems[29]; though these data were chosen to provide a baseline measure that aligned with the questions in the survey, the health information system (HIS) indicators (ie, # of women receiving >4 ANC check-ups, and the number of women given iron and folic acid tablets) conflate demand-side and supply-side factors. The study provided the proportion of women provided ANC services at T1 and T2 (as a measure of changes in service provision), and the number of women receiving services at T0. The study found that ANC services were

disrupted during the lockdown period but were restored during the post-lockdown period, but not to prepandemic levels).[29]

The study from Bangladesh included a baseline assessment carried out in early 2020 (T0), and follow-up assessments during lockdown (T1) and post-lockdown (T2), asking healthcare workers about their exposure to training, their workload and time commitments, and the types of services (specifically health and nutrition services for mothers and young children) they were providing (eg, was service X provided in T1).[26] Data were also collected from pregnant women and mothers of young children (eg, was service X received) at the same time points. The study provides a proportionate measure of service provision (defined as the proportion of service providers indicating provision of specific services, and the proportion of service users indicating receipt of specific services) at prepandemic (T0), lockdown (T1) and post-lockdown (T2). The study found that fewer facilities offered ANC services during lockdown when compared with before the pandemic (dropping 6.6 percentage points), and that there were significant drops in the receipt of anthropometric measurements, and iron/folic acid/calcium supplementation. All services recovered post-lockdown but not to prepandemic levels.[26]

### Surveys of healthcare workers to construct composite service disruption scores

Surveys of healthcare workers to construct composite service disruption scores for specific services (including MNH services) were used in Burkina Faso, Ethiopia, and Nigeria.[14] Responses were scored 0 (no interruption), 1 (partial interruption), 2 (complete interruption) for each of the three MNH services (ie, ANC, iron and folate supplementation, and family planning services) for a maximum disruption score of 6. The average of all individual responses was calculated for each of the three study settings. Maternal and reproductive services were scored an average of 2.24 (out of 6) for Nigeria, 1.72 for Burkina Faso and 1.67 for Ethiopia.[14]

### Survey of service users to measure receipt of health services

A survey of service users to measure receipt of health services was carried out in Iran to assess the actual percentage of services delivered.[38] An SMS message (ie, 'Has the service been received?' [yes/no]) was sent to all patients attending hospital services. Data were collected before and during the pandemic and were aggregated by calendar month. The study indicates the proportion of attendees who received a service, by month (eg, the study found that there was a small but significant drop in the 'actual percentage of service delivery' for midwifery services when comparing the pandemic to prepandemic periods).[38]

### Observation of health facilities

Observation of health facilities was carried out in Nepal, using a structured clinical observation checklist,

to determine the number of healthcare workers in the labour and delivery room, the number of workers redeployed to COVID-19, and the number of days of disruption of maternity services (ie, no maternity services provided on these days).[39] The study collected baseline data over 6 months in 2019 (prepandemic period), followed by data collection over 6 months in 2020 (pandemic period). The study measured the mean difference in the number of health workers in the labour room, and the number of days that maternity services were disrupted between the two time periods. The study provided a mean number of days (4.3) that maternity services were halted during the pandemic period (March–August 2019), and concluded that the number of healthcare workers (per 24 hours in the labour and delivery room) had decreased to 5.4 (from 6.2 before the pandemic) due to redeployment to COVID-19-dedicated care.[39]

### Qualitative measures of disruption

Various qualitative approaches were used to generate accounts of disruptions to the provision or quality of services including: (1) semistructured interviews with service providers and/or service users to elicit descriptions of experiences of disruption,[28 30 31 33 34 36 37 39–46] (2) semistructured interviews with service providers eliciting comparisons of the provision of care at different times,[35 49] (3) semistructured interviews among service users asking about management, barriers, and facilitators influencing service delivery (stratified by levels of health system),[47] (4) free-text field in a survey of service providers,[50 51] and (5) media analysis of news reports.[32] Some of these sources focused on the experiences of respondents during a lockdown period,[34] or during a lockdown period compared with the period preceding or following lockdown.[33 42] Only six of the 21 qualitative (or mixed-methods) studies published the original data collection instrument.[31 34 41 43 47 51]

### Provision of services

Disruptions to provision of services included: closed health facilities or village health posts,[30 39] facilities that had closed certain wards/units/clinics/services (eg, outpatient clinics,[34 44 50] doctor's offices,[37] support for kangaroo mother care,[40] counselling for care of mother and newborn[40]), limits to the number of patients that could be seen,[36 46] lack of follow-up[46 50] or home visits,[35 43] redeployment of staff to support COVID-19 services,[32 39 43 50] reduced beds or space,[43] reduced hours of service,[46 47] health facilities converted to COVID-19 care centres,[33 47] unavailable healthcare workers,[33 47] lack of laboratory services,[30] and specific services (eg, intranatal services) unavailable[31] or reduced.[31 35 42 43 47]

### Quality of services

Disruptions to the quality of services included: poor quality of care,[32 33 37 40 46 47 49] long waiting times,[33 36 45] poor attitude of healthcare workers towards clients,[28] reduced numbers of healthcare workers,[39 40 44 47] reduced physical

and/or emotional support,[49] staff preoccupied with infection prevention and control,[46] unhygienic conditions,[47] shorter times with healthcare workers,[37 40 47] healthcare workers unwilling to perform certain examinations/procedures (eg, abdominal palpation, auscultation of foetal heart rate),[37 47] increased risk of medically unjustified caesarean section,[49] and shortages of resources (including medicines, vaccines and/or oxygen).[28 37 47 51]

## DISCUSSION

Our initial screening (on title and abstract) revealed that the majority of the literature on service disruption reports on demand-side measures. This was also a finding of a recent review of health system performance indicators.[53] Our search identified a small number of measures isolating supply-side factors (N=28). Among these, nine sources included quantitative measures, of which two focused on the causes (as opposed to the presence or extent) of disruption.[27 52] The 21 qualitative (including 2 mixed-methods) studies included rich descriptions of the experiences of supply-side delay from the perspective of both service-providers and service users, yet none of the findings from these studies appear to have informed the design of the quantitative surveys. Only one study used empirical findings (from a previous iteration of the study survey) to pre-populate survey fields describing the nature of service disruption.[52] We believe the qualitative studies may provide important contextual relevance and granularity to future surveys.

Emergency preparedness guidance documents often emphasise the importance of monitoring and evaluation but typically do not include indicators of health system disruption. The WHO's COVID-19 Strategic Preparedness and Response Plan states, '...it is important to track health system capacity and performance; including hospital and intensive care bed occupancy rates, as well as the health system's ability to continue providing non-COVID-19 services'; however, a bank of specific indicators is not included.[54] The evidence we identified suggests that approaches to measuring supply-side disruption are varied, inconsistent, and largely *ad hoc*. We identified a small number of measures of supply-side disruption that included real-time baseline measurements; this suggests that systems can be put in place at the onset of a pandemic to allow for the collection of baseline measurements. This is particularly important as retrospective baseline measurements often lack precision as they are typically prone to recall and confirmation biases. In addition, early planning for periodic follow-up is important as, '...metrics of system responses and resilience may only be understood appropriately as we relate them to the severity and dynamics of the shock itself' (Fleming et al,[55] p1202).

Facility-level assessments (and community-level assessments) designed to establish the continuity of essential services in the context of COVID-19 are important tools for monitoring the provision and quality of care; however,

they may require adaptation for future outbreaks (and other crises) and they may not be suitable for specific health programmes or priorities.[5 56] In addition, our review suggests that there are additional approaches to measuring supply-side disruption that may provide an important complement to facility assessments. The evidence also suggests that qualitative interviews may be a more suitable approach to collecting data about quality of services and may provide the opportunity for participants to offer localised solutions. Qualitative methods may also clarify the mechanisms through which services are affected (eg, including cascading disruptions, proxy measures of disruption) which may help to identify targeted mitigation strategies.[53]

Finally, we excluded one source reporting on a clinical review of patient records to assess the extent of unnecessary hospitalisations of pregnant women in Tajikistan.[57] This study was excluded as it was focused on demand-side indicators (ie, total hospitalisations before and during COVID-19) and only assessed unnecessary hospitalisations for two months during the pandemic as a measure of the strength of the existing primary healthcare system. However, though the indicator was not suitable for assessing COVID-19-related disruption in this context, we believe that unnecessary hospital admissions (and unnecessarily prolonged admissions) could be a suitable indicator of disruption to quality of primary healthcare—insofar as it reflects poor organisation, poor use of resources, and reduced inpatient capacity—if collected prospectively, or retrospectively to include the preoutbreak period. Ultimately, this example highlights that service uptake may function as a proxy indicator of quality; however, service uptake alone is not suitable as a robust measure of the quality of care provided or received.

### Limitations of the evidence
Many of the included sources provided weak description of methods. Some sources aimed to measure supply-side disruption but presented crude measures of provision (eg, number of ANC visits provided per day) which may conflate supply and demand-side disruption. While the qualitative studies provide important context-specific information, they may not be transferable to other settings. Many respondents indicated that services were closed or that they were of low quality during the pandemic (or specifically during lockdown); however, few sources aimed to determine to what extent services were available, and at what comparative quality, prior to the pandemic. Concerns are often raised about the pandemic undermining the steady gains made in the provision of MNH services in recent years[10]; thus, comparing disruption to a baseline measure fails to account for gains that would have been made otherwise and may, therefore, underestimate the true extent of the disruption.

### Limitations of the review
While COVID-19-related disruption was globally pervasive, there are relatively few sources focused on

the disruption of MNH services. As service disruption may have been assumed to be unavoidable (and thus not worthy of measuring), it may be underreported. Similarly, acknowledging disruption may be viewed as a liability, or may be considered demoralising to service providers, which may have resulted in a reluctance to measure disruption, or disseminate disruption data. We opted for a broad search strategy to avoid missing sources that did not specifically reference 'disruption'; however, we still may have missed relevant sources which did not reference 'service' or 'programme' in the title or abstract. In addition, we suspect that publication delay may have limited the amount of available evidence. Finally, we were unable to assess the quality of the evidence due to gaps in reporting across nearly all the identified studies.

## CONCLUSION

The evidence suggests that: (1) there is an overrepresentation of demand-side indicators, (2) the methods for isolating supply-side disruption are varied and inconsistent, (3) there are few published studies of supply-side disruption that include baseline data (with most relying on retrospective baseline estimates or subjective estimation of change over time), (4) relatively few of the quantitative indicators are designed to measure the *causes* of disruption, (5) there are few studies reporting on supply-side disruption from the perspective of service users, and (6) there is a lack of a clear or nuanced (ie, including both the provision of services and quality of services) framework for defining supply-side disruption. Furthermore, there are few studies which measure disruption across all essential services. While in some settings MNH services were reported to be less disrupted than other essential services, it is still important to measure disruption across all essential services to ensure equitable scale-down in response to supply-side constraints.

Taken together, the evidence emphasises the importance of monitoring essential health services and highlights the need for specific guidance for defining service disruption, and for developing indicators that are able to isolate supply-side disruption, both to the provision and quality of services. Given the difficulty of setting up robust monitoring systems during an emergency, and given the value of baseline measures and regular follow-up to identify the early signs of disruption, we recommend that strategies for measuring disruption be developed and incorporated into preparedness plans. Finally, we suggest that preparedness plans provide specific indicators—including indicators for measuring disruption to MNH services—alongside guidance on how data should be collected, and the resources required to support data collection and reporting in an emergency.

## OTHER

The review protocol has been published in the PROSPERO prospective systematic review registry (CRD42022381537).

**Author affiliations**
[1]Department of Infectious Disease Epidemiology, London School of Hygiene & Tropical Medicine, London, UK
[2]Leeds Teaching Hospitals NHS Trust, Leeds, UK
[3]Humanitarian Department, Save the Children International, London, UK
[4]Croydon University Hospital, Croydon, UK
[5]Department of Humanitarian Response, Save the Children Federation, District of Columbia, Washington, USA
[6]Department of Publiic Health & Policy, London School of Hygiene & Tropical Medicine, London, UK

**Acknowledgements** We would like to thank Dr Louisa Baxter for providing guidance on clinical and programming quality. Thank you to Dr Alicia Renedo for coding the Spanish language source. Thank you also to Dr Shirley Huchcroft for her invaluable feedback on an early draft of this paper.

**Contributors** Conceptualisation: CRM/EM/LJC. Formal analysis: CRM/NA/LR/MG/JM. Funding acquisition: LJC. Guarantor: CRM. Methodology: CRM/DG. Project administration: LR/JM/LJC. Supervision: EM/JM/RC/LJC. Writing (original draft): CRM. Writing (reviewing and editing): CRM/DG/EM/NA/LR/MG/JM/RC/LJC.

**Funding** This review was made possible by the generous support of the American people through the US Agency for International Development (USAID).

**Disclaimer** The contents of this manuscript are the responsibility of READY Initiative and do not necessarily reflect the views of USAID or the United States Government.

**Competing interests** None declared.

**Patient and public involvement** Patients and/or the public were not involved in the design, or conduct, or reporting, or dissemination plans of this research.

**Patient consent for publication** Not applicable.

**Provenance and peer review** Not commissioned; externally peer reviewed.

**Data availability statement** All data relevant to the study are included in the article or uploaded as online supplemental information. Not applicable.

**ORCID iDs**
Catherine R McGowan http://orcid.org/0000-0001-6941-6539
Meghan C Gallagher http://orcid.org/0000-0001-7572-962X

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
