## [Reviewer comments · BMJ Open]

This paper was submitted to a another journal from BMJ but declined for publication following peer review. The authors addressed the reviewers' comments and submitted the revised paper to BMJ Open. The paper was subsequently accepted for publication at BMJ Open.

ARTICLE DETAILS

TITLE (PROVISIONAL)	Measuring supply-side service disruption: A systematic review of the methods for measuring disruption in the context of maternal and newborn health services in low- and middle-income settings
AUTHORS	McGowan, Catherine; Gokulakrishnan, Dhikshitha; Monaghan, Emily; Abdelmagid, Nada; Romig, Laura; Gallagher, Meghan; Meyers, Janet; Cummings, Rachael; Cardinal, Laura

VERSION 1 – REVIEW

REVIEWER	Garfield, Richard CDC
REVIEW RETURNED	03-Sep-2023

GENERAL COMMENTS	The topic is important and the methods are appropriate, but the results are quite thin. Authors have, in effect, a small grab bag of diverse methods that in the end fail to assess adequately the contraction of services during COVID. Thus appropriate though the methods re, the research is not very useful. Note: Funders had detailed information on the contraction of services, especially the Global Fund that used this info to disburse additional funding. Our knowledge from this and from epi information on mortality and reportable diseases do give us a rich picture of service contraction and its effects. You would not know that from this paper.
--

REVIEWER	Maliqi, Blerta World Health Organization, Maternal Newborn Child Adolescent Health and Ageing
REVIEW RETURNED	09-Sep-2023

GENERAL COMMENTS	I would like to thank the co-authors for having attempted to analyse a critical question such as that of quality of care for MNCH services during COVID pandemic, from a supply side. I found the methods, analysis and conclusions of good quality. However, I would like to draw the attention of the co-authors that while their analysis from a supply perspective is not wrong, their contextualisation of "supply" and use of "supply indicators" in pages 6-11, often slips towards use of indicators that describe utilisation of services. Utilisation is a composite measure that is used to summarise both, supply and demand issues, not supply issues alone. E.g. Page 6, lines 22-27 use utilisation as supply
---

	measure (correct use would "utilisation indicators as a proxy of supply", similarly page 8 lines 8-9 made similar assumption. Also, utilisation alone is a proxy measure of quality but not a robust measure of quality of care given or received. The table of indicators attached as Annex 2 is far more clear and close to the type of supply indicators linked directly to quality of care than those used in the first part of the paper (as a proxy to understand the "supply side of quality of care"). I would encourage authors to use those more upfront, rather than leave it to the reader to check (or not) the annex. Also, I would encourage the authors to refer to "Analysing and using routine data to monitor the effects of COVID-19 on essential health services: practical guide for national and subnational decision-makers" https://www.who.int/publications/i/item/WHO-2019-nCoV-essential_health_services-monitoring-2021.1 which can give them a stronger entry point to articulate their question and anchor their conclusions. Overall, I recommend the authors correct the use of "utilisation indicators" in the introduction and methods section, and give more clear examples or a reference to the framework used to define quality of care from a supply side of MNCH services. WHO's standards on delivery quality care for maternal and newborn health in health facilities, and the related standards on small and sick newborn, and pediatric care, would be a good solution to the paper framing problem.
--	--

VERSION 1 – AUTHOR RESPONSE

Reviewer 1 comments

Reviewer 1 comment	Authors' response
The topic is important and the methods are appropriate, but the results are quite thin. Authors have, in effect, a small grab bag of diverse methods that in the end fail to assess adequately the contraction of services during COVID.	We are grateful to Reviewer 1 for taking the time to review our manuscript. However, this comment is most puzzling as "assessing the contraction of services during COVID-19" was not an objective of the review. The review sought to present the methods used, in practice, during COVID-19, to measure disruption as currently (or when we submitted the manuscript at least) there were no available guidelines on measuring disruption. The reason we summarised the qualitative findings is simply to inform the design of quantitative indicators. Were we tasked with coming up with indicator guidance this is the information we would need. To clarify our intentions we have added the following to the end of the Rationale subsection in the Introduction: "Furthermore, the design of indicators is typically evidenced by qualitative research findings; as such, we sought to consolidate qualitative findings

	regarding the nature and extent of service disruption”.
Thus appropriate though the methods re, the research is not very useful.	We defer to Reviewer 2’s assessment of our manuscript’s utility and reiterate that were Reviewer 1 under the impression that the authors sought to consolidate the evidence for disruption – rather than consolidate the methods for measuring disruption – the manuscript would indeed not be terribly useful. Regardless, this was not an objective of the review.
Note: Funders had detailed information on the contraction of services, especially the Global Fund that used this info to disburse additional funding.	We were not privy to this information, nor did it emerge in our research in a relevant manner. Thus, we did not incorporate it into this manuscript.
Our knowledge from this and from epi information on mortality and reportable diseases do give us a rich picture of service contraction and its effects. You would not know that from this paper.	As noted above, assessing the contraction of services during COVID-19 was not an objective of the review. The review presented the methods used, in practice, during COVID-19, to measure disruption as, at the time of writing, there were no available guidelines on measuring disruption. We appreciate the feedback from Reviewer 1 and hope that we have clarified sufficiently to address your concerns.

Reviewer 2 comments

Reviewer 2 comment	Authors’ response
I would like to thank the co-authors for having attempted to analyse a critical question such as that of quality of care for MNCH services during COVID pandemic, from a supply side.	We thank Reviewer 2 for this comment and for taking the time to review our manuscript. Reviewer 2’s comments are excellent and provide the opportunity to make important improvements to our paper.
I found the methods, analysis and conclusions of good quality.	Thank you.
However, I would like to draw the attention of the co-authors that while their analysis from a supply perspective is not wrong, their contextualisation of "supply" and use of "supply indicators" in pages 6-11, often slips towards use of indicators that describe utilisation of services. Utilisation is a composite measure that is used to summarise both, supply and demand issues,	With respect to the comment about p. 6 we were not entirely sure which p. 6 was being referenced but have changed the wording on p. 1 (at the bottom of the page) to: “Demand-side indicators may function as proxy indicators of supply insofar as they may evidence a reluctance to present to health services, or challenges with access”. And we have clarified on p. 2 as follows: “Both the WHO guidance on maintaining essential health

not supply issues alone. E.g. Page 6, lines 22-27 use utilisation as supply measure (correct use would "utilisation indicators as a proxy of supply", similarly page 8 lines 8-9 made similar assumption.	services, and the guidance on using routine data to monitor the effects of COVID-19 on essential services, include lists of sample indicators for measuring disruption; however, most of these indicators measure demand-side disruption as a proxy for supply-side disruption (e.g. number of women presenting to facility), or as a composite measure of both supply- and demand-side disruption (e.g. number of facility births) (4, 6)".
Also, utilisation alone is a proxy measure of quality but not a robust measure of quality of care given or received.	This is an excellent comment. As our discussion of an excluded paper seeking to measure unnecessary admissions is perhaps a good example to highlight this fact we have chosen to add the following at the end of this section: "Ultimately, this example highlights that service uptake may function as a proxy indicator of quality; however, service uptake alone is not suitable as a robust measure of the quality of care provided or received".
The table of indicators attached as Annex 2 is far more clear and close to the type of supply indicators linked directly to quality of care than those used in the first part of the paper (as a proxy to understand the "supply side of quality of care"). I would encourage authors to use those more upfront, rather than leave it to the reader to check (or not) the annex.	Thank you for your positive feedback regarding Annex 2. To better incorporate these indicators into the body of the manuscript, we've revised a portion of the Eligibility Criteria within the Methods Section. We've used indicators from Annex 2 as examples for supply side disruption in the context of the provision of care and the disruption that compromises the quality of care.
Also, I would encourage the authors to refer to "Analysing and using routine data to monitor the effects of COVID-19 on essential health services: practical guide for national and subnational decision-makers" https://www.who.int/publications/i/item/WHO-2019-nCoV-essential_health_services-monitoring-2021.1 which can give them a stronger entry point to articulate their question and anchor their conclusions.	Thank you for this resource. We have added the following to the Rationale section: "Whilst we were able to identify useful guidance on monitoring the effects of COVID-19 on essential health services, this guidance focussed exclusively on routinely collected data (17). Importantly, the authors caution that routine health information systems (RHIS), "...present an incomplete picture of the services used by communities because reporting rates are often low and many private care facilities (including those run by nongovernmental or religious organizations) do not report data to RHIS" (17, p. 5). In addition, RHIS present only a partial indication of the impact of COVID-19 on essential services in settings in which a significant proportion of health services are provided through the private sector (17). Finally, RHIS data vary in terms of data quality and completeness, and population denominators are particularly prone to error. This review aims to address existing gaps by

	describing how COVID-19-related, supply-side disruption has been measured in the context of MNH services in LMICs during COVID-19.”
Overall, I recommend the authors correct the use of "utilisation indicators" in the introduction and methods section, and give more clear examples or a reference to the framework used to define quality of care from a supply side of MNCH services. WHO's standards on delivery quality care for maternal and newborn health in health facilities, and the related standards on small and sick newborn, and pediatric care, would be a good solution to the paper framing problem.	Thank you for your feedback. We have corrected the use of “utilisation indicators” throughout per your recommendation above. We have also included a reference to, and examples from, the WHO standards in the Eligibility Criteria section to add nuance to our definition of quality.

VERSION 2 – REVIEW

REVIEWER	Maliqi, Blerta World Health Organization, Maternal Newborn Child Adolescent Health and Ageing
REVIEW RETURNED	14-Nov-2023

GENERAL COMMENTS	Thank you for having considered the feedback. Paper is stronger and reads better. The framework suggested for the analysis of the paper was the one used for the MNCH standards: Provision of care (Supply) and experience of care (demand driven by both experience of users and experience of care providers), underpinned by the health systems inputs and processes. Also, the AAAQ framework (based on Tanahshi) could have helped. However, the paper stands as it and it is worth publishing it, Congrats to the authors.
--